# Biomedical Applications of Electromagnetic Detection: A Brief Review

**DOI:** 10.3390/bios11070225

**Published:** 2021-07-07

**Authors:** Pu Huang, Lijun Xu, Yuedong Xie

**Affiliations:** 1School of Instrumentation and Optoelectronic Engineering, Beihang University, Beijing 100191, China; huang_pu@buaa.edu.cn; 2Beijing Advanced Innovation Centre for Big Data-Based Precision Medicine, Beihang University, Beijing 100191, China; lijunxu@buaa.edu.cn

**Keywords:** electromagnetic detection and biosensors, electromagnetic biological theory, biomedical application, frequency, machine learning

## Abstract

This paper presents a review on the biomedical applications of electromagnetic detection in recent years. First of all, the thermal, non-thermal, and cumulative thermal effects of electromagnetic field on organism and their biological mechanisms are introduced. According to the electromagnetic biological theory, the main parameters affecting electromagnetic biological effects are frequency and intensity. This review subsequently makes a brief review about the related biomedical application of electromagnetic detection and biosensors using frequency as a clue, such as health monitoring, food preservation, and disease treatment. In addition, electromagnetic detection in combination with machine learning (ML) technology has been used in clinical diagnosis because of its powerful feature extraction capabilities. Therefore, the relevant research involving the application of ML technology to electromagnetic medical images are summarized. Finally, the future development to electromagnetic detection for biomedical applications are presented.

## 1. Introduction

Recently, the biological detection technology has developed rapidly. It can be used to detect the relevant parameters that characterize the performance of the organism due to their unique identification ability to the organism [1]. When the organism is stimulated externally, the organism converts these stimulation signals into electrical signals that can be received and processed. This attribute allows it to obtain nutrients or stay away from danger. Human beings use the sensitivity of biometrics to observe and understand the living environment. In other words, the core of biological detection is converting observable things into measurable physical quantities as the result of biological simulation. In fact, biological detection technology has made greater breakthroughs in bio-electric signal conversion because of the continuous integration and penetration of biotechnology, chemistry, biology, physics, electronics, and information technology [2,3,4]. According to the detection mechanism, biological detection technology can be divided into electromagnetic biological detection technology, electrochemical biological detection technology, optical biological detection technology, semiconductor biological detection technology, thermal biological detection technology, piezoelectric crystal biological detection technology, etc.

Compared with other detection technology, electromagnetic biological detection technology has received attention from an increasing number of researchers due to the characteristics of high sensitivity and high detection speed [5]. The basic principle of electromagnetic detection technology is electromagnetic biological effect. Specifically, organisms can produce biological tissue effects when they are affected by the external electromagnetic field. People can get relevant information based on the relevant reactions of organisms. Since the inextricable connection between biological electromagnetic effects and life systems, it can be used to explore the positive effects of electromagnetic fields to promote the development of humanity.

More important, the biological action mechanism and biological effects of electromagnetic fields are closely related to the excitation frequency, intensity, and duration of electromagnetic fields [6]. Different electromagnetic fields have different biological targets, scope of action, action time, and energy and information exchange methods [7]. Therefore, electromagnetic detection has a wide range of applications. For example, electromagnetic biosensors have played an important role in increasing the germination rate of seeds [8], improving the adaptability of organisms to the environment [9], controlling environmental pollution [10], etc. At the same time, bioelectromagnetic detection is also applied in body health monitoring, especially the emergence of nuclear magnetic resonance technology, which has pushed bioelectromagnetic detection to a new peak. In addition, treatment methods based on electromagnetic biological effects have emerged one after another, such as magnetized water and magnetic acupuncture. The study of bioelectromagnetic detection is of great significance to promote the level of social medical care and improve the quality of human life.

This article mainly introduces the biomedical applications of electromagnetic detection in recent years. Specifically, this review first introduces the electromagnetic biological effects and their mechanism of action. In fact, the excitation frequency and intensity are the main parameters affecting electromagnetic biological effects. Taking the excitation frequency of the electromagnetic field as a clue, the application of bioelectromagnetic detection, including health monitoring and disease treatment, is discussed. Furthermore, electromagnetic detection in combination with ML technology has been used in clinical diagnosis since the ML can extract the deep features of electromagnetic signals to obtain deep biological information. The relevant research involving the application of ML technology to electromagnetic medical images is summarized. Finally, the author presents a conclusion on the reviewed biosensing techniques, and provides an opinion on the future development trend of electromagnetic detection.

## 2. Electromagnetic Biological Effect and Mechanism

When the electromagnetic field acts on an organism, various substances in the organism will react to the alternating electric and magnetic fields in the way of induction, which in turn results in the biological effect of the electromagnetic field. The biological effects of electromagnetic fields can be divided into thermal effects, non-thermal effects, and cumulative effects [11]. The biological thermal effect of electromagnetic wave refers to the biological effect caused by the Joule heat converted from the electromagnetic wave penetrating the biological system [12]. Compared with thermal effects, the non-thermal effects are changed in tissues or systems, that is, there is no direct relationship with heat after the organism absorbs electromagnetic radiation energy [13]. In addition, if an organism damaged by thermal and non-thermal effects is not repaired in time, the degree of damage will accumulate when it is exposed to electromagnetic radiation again. The above process is called the cumulative effect.

A series of complex bioelectrical activities are regarded as an important part of the whole organism. The shape, structure, and function of different biological levels will change when a certain intensity of electromagnetic waves penetrate the biological system. In fact, the biological action mechanism is closely related to the frequency, intensity, and duration of electromagnetic fields [14]. The scope of action of biological objects, action time, energy and information exchange methods, and exchange values are all different due to the different characteristics of electromagnetic waves, which brings different applications. Table 1 summarizes different electromagnetic effects and their applications. In order to analyze and understand the electromagnetic biological effect, it is necessary to summarize and sort out the hypothesis of the electromagnetic biological effect mechanism.

### 2.1. Biological Thermal Effects

Thermal effect refers to the effect produced by electromagnetic radiation heating biological tissues or systems. The strength of the thermal effect is monotonously and positively related to the amount of electromagnetic power absorbed by the biological system. There are three ways to generate the thermal effect, namely, friction heat generation, dielectric loss heat generation, and conduction current heat generation [25].

Friction heat generation is caused by the redistribution of electric charges of water molecules, amino acids, DNA, and other biological molecules under the action of alternating electromagnetic fields [26]. It leads to changes in the structure and function of biomolecules. If the frequency of the alternating electromagnetic field is high, it will have two effects. On the one hand, various polar molecules in the biological system will undergo rapid and periodic changes. In the process of change, these polar molecules collide and rub against the surrounding molecules violently, which generates a large amount of heat energy. On the other hand, various ions in the biological system undergo periodic motion near their equilibrium position due to the action of alternating electromagnetic fields. During the movement, these ions collide with other molecules to generate heat energy.

There are positive and negative charges in the molecule. Under the action of the electric field, these positive and negative charges are subjected to electric field forces in different directions, resulting in corresponding displacements [27,28]. The coupling dipole moment and polarization direction of polar molecules changes due to charge displacement, which may lead to changes in the molecular structure. In fact, it takes time to change the polarization of the dielectric. More specifically, if the electric field changes faster than the polarization direction, the polarization of the dielectric will not keep up with the electric field change, and the electric field energy will be lost.

The last electromagnetic heating effect is conduction current heat generation. There are charged particles that can migrate freely in organisms, which makes the organisms generate an electric current under the action of an electric field; the conduction current passing through the tissue can generate heat. The generation of thermal effects generally requires a higher intensity electromagnetic radiation. At present, the research on the thermal effect mechanism is relatively clear.

### 2.2. Biological Non-Thermal Effects

There are weak electromagnetic fields in the organs and tissues of organisms. These electromagnetic fields are stable and orderly. The non-thermal effect of electromagnetic field will interfere with the inherent weak electromagnetic field inside of the organism. Once the inherent electromagnetic field in a balanced state is disturbed by the external electromagnetic field, the tissues and organs in the organism are damaged. Compared with the electromagnetic thermal effect, the non-thermal effect has four main characteristics. The first one is coherence, that is, the biological effect can only be produced when the electromagnetic wave parameters and the relevant parameters of the biological system meet a certain deterministic relationship. The second feature is the window characteristics, which mainly consists of frequency windows and intensity windows. The former refers to the effect that the target in the biological system only produces certain discrete electromagnetic waves with a very narrow frequency range. The latter refers to the biological effect that the target in the biological system only produces on certain discrete electromagnetic waves with extremely narrow field strength. Another feature is synergy. The weak electromagnetic field and the metabolism of the biological system can stimulate extremely strong biological effects. The last feature is non-linearity. The intensity of the effect will not increase significantly when the intensity of the electromagnetic wave is greater than a certain threshold. Non-thermal effects can occur under extremely low frequency electromagnetic fields and low-intensity radio frequency radiation. In recent years, the non-thermal effects of electromagnetic fields on biological systems have become a research hotspot due to the extensive application of power equipment and radio frequency radiation.

#### 2.2.1. Coherent Oscillation Theory

Regarding the mechanism of the non-thermal effects of electromagnetic fields, Frohlich first proposed the coherent oscillation theory [29]. The theory believes that biological molecules with dipole characteristics have a fixed vibration frequency due to the interaction of the Coulomb gravitational force. Therefore, the polarized wave is formed, and the coherent effect is produced. When the external electromagnetic field energy is introduced into this fixed vibration, it will be stored in the oscillation mode in a purely mechanical manner. The biomolecules are strongly excited and stay away from the thermal equilibrium state. If the external electromagnetic field close to the coherent oscillation frequency of the biological system acts on the biological system, it will produce two kinds of “frequency window” effects. These two kinds of effects are destructive coherence and constructive coherence, which leads to obvious biological reactions.

#### 2.2.2. Ion Transmembrane Cyclotron Resonance Theory

The theory of ion transmembrane cyclotron resonance is another theory referring the non-thermal effects. The theory considers that charged ions in a static magnetic field move in a circular motion under the action of the Lorenz force [30,31]. In the electromagnetic radiation, the alternating magnetic field parallel to the static magnetic field will increase the angular velocity and orbit radius of the charged particles, which in turn produce cyclotron resonance. When the cyclotron resonance frequency is consistent with the ELF frequency of the intracellular ${\mathrm{Ca}}^{2+}$, it will induce the receptors on the cell surface and the ions in the transmembrane channel to move circularly or spirally. This phenomenon will affect many enzymes regulated by calmodulin and induce various physiological and biochemical changes. At the same time, the electromagnetic field changes the dipole moment of the ion channel, and the cyclotron motion of the ion in the channel will eventually interfere with the process of ion permeability.

#### 2.2.3. Free Radical Mechanism Theory

The currently commonly recognized mechanism is the free radical pair mechanism [32]. According to the quantum theory, the chemical bond formed between two reactive free radicals is requested to be in a single state. Since electrons have a magnetic moment, the local magnetic field generated by other electrons or atomic nuclei in the molecule may reverse the spin direction of an electron. The radical pair changes from a single state to a triplet state. More important, the external magnetic field will maintain the magnetic moment of the electrons and reduce the possibility of spin reversal, thereby affecting the reaction rate. Although the experiment shows that the reaction exists in biological systems, its biological significance is not yet fully defined [33].

### 2.3. Cumulative Effects

It takes a certain time for the magnetic field to change the metabolism, function, and structure of the organism. This phenomenon is called hysteresis. The main reason for the hysteresis is the existence of various dipoles, ion genes, and inorganic ions in the biological macromolecule [34]. They are restricted by steric hindrance and potential energy. In addition, only a few stable equilibrium positions are available for the particles in space. They can jump from one equilibrium position to another under the effect of external electric fields. There is also an electric dipole moment between each other because ionic groups and complex ions exist in the form of different chemical bonds. When the electric dipole moment rotates towards the outward electric field, a certain potential barrier must be overcome, so there must be a time course. The electric dipole moment must overcome a certain barrier to rotate towards the external electric field, which must take a certain amount of time. From the energy point of view, a higher the product between the field strength and the action time increases the biological effect, which reflects the cumulative process.

## 3. Application of Electromagnetic Detection Technology at Different Frequencies

The utilization of electromagnetic biological effects involves many aspects, including disease treatment [35], food quality inspection [36], food sterilization [37], plant cultivation [38], and so on. Since the main parameters affecting the electromagnetic biological effects are frequency and intensity, electromagnetic waves at different frequencies produce different biological effects. According to the interaction mechanisms between biological tissues and electromagnetic fields at different frequencies, the radiated electromagnetic fields are usually divided into four frequency bands: electrostatic field (0 Hz), extremely low frequency (ELF, 0–300 Hz), intermediate frequency (300 Hz–10 MHz), and radio frequency (RF, 10 MHz–300 GHz). The frequency band of the electromagnetic field is shown in Figure 1. Therefore, this review discusses the related application of electromagnetic biological detection by using their working frequency band as a clue.

### 3.1. High-Voltage Electrostatic Field (0 Hz)

In recent years, high-voltage electrostatic fields have been widely applied in agricultural production and food processing. This technology has an effect on water molecules, enzymes, and other substances inside crop seeds to optimize their original properties and achieve good conditions for seed growth. When the seeds are treated with high-voltage electrostatic fields, ion mist, $O_{3}$, and nitrogen oxides will be produced. Nitrogen oxides on the surface of the seed compound with hydration to form nitric acid and nitrous acid, which can stimulate embryos in the dormant state of the seed. Moreover, the $O_{3}$ is a strong bactericide, which can greatly reduce the incidence of plant diseases. Therefore, the application of electrostatic field helps to obtain high-yield, high-quality, and non-toxic agricultural products [39]. As for the aspect of food processing, the high-voltage electrostatic field technology can not only be used for the fresh-keeping of fruits and vegetables, but also in food sterilization and thawing to meet human dietary needs.

#### 3.1.1. Seed Germination and Growth

The earliest and most extensive field of electrostatic biological effects research is the use of high-voltage electrostatic fields to treat plant seeds. In 1962, Krueger et al. first discovered that the air environment ionized by positive or negative ions can significantly boost the germination rate of oat seeds [40]. In addition, the growth rate and dry weight of oat seeds after germination are also significantly improved. The experiment equipment is shown in Figure 2 [41]. Sidaway et al. placed lettuce seeds between energized aluminum electrodes for 24 h [42]. The results showed that both positive and negative electrostatic fields from 180 V/cm to 360 V/cm can change the germination rate of lettuce seeds. Figure 3 is the result of high-voltage electrostatic field treatment of safflower seeds, which further proves the effectiveness of the technology [43]. More important, they found that the positive electric field has an inhibitory effect on plant growth, while the negative electric field has a promoting effect. Huang R. et al. treated cucumber seeds with swelling treatment and electric field treatment, respectively, and then carried out related experiments about germination and membrane permeability [44]. The results showed that these two treatment methods can improve the germination rate of cucumber seeds. In subsequent studies, it was further discovered that the combination of a high-voltage electrostatic field and magnetic field technology has a more significant effect on increasing the germination rate of plants. Based on this technology, Li et al. applied a high-voltage electrostatic field with an intensity of 2.25–2.5 kv/cm to tomato seeds, while applying an alternating magnetic field over 8 h [45]. It was found that the germination rate of tomato seeds was increased by 1.1 to 2.8 times compared to the control group. Shao applied Helmholtz coils with the intensity of 2 A to induce the magnetic field. The magnetic field with discharge plasma can improve the germination rate and activity of old spinach seeds. In fact, the germination rate and activity of seeds depend on the flux of the excitation magnetic field [46]. Xu et al. applied corona discharge produced by a multi-needle-plate HVEF to treat naked oat seeds. According to the Fourier Transform infrared spectroscopy (FTIR) of the seed coat, high-voltage electrostatic fields treatment can form a new absorption peak at 1740 cm^−1^, which is closely related to the hydrophilicity of the seed; that is to say, the high-voltage electrostatic fields treatment can improve the hydrophilicity of seeds [47].

A high-voltage electrostatic field not only improves the germination rate of seeds, but also improves the physiological indicators of seedlings. Kiatgamjorn et al. found that the germination rate of seeds irradiated with electric fields of 10 kv/m and 25 kv/m are improved, and the seedling height and root length of seedling are significantly higher than seedlings of the control group [48]. In addition, the effect of high-voltage electrostatic field on plant seeds is also manifested in improving the germination index, vigor index, and the activity of various enzymes. Isobe et al. used hydrogen spectrum technology to find that high-voltage electrostatic field treatment can enhance the combined water content of morning glory seeds [49]. In fact, different plants have different sensitivities to a high-voltage electrostatic field environment, and the unsuitable electric field environment may have a negative effect on seed germination. Although Isobe et al. applied a high-voltage electrostatic field of suitable strength to increase the germination rate of morning glory seeds, the germination rate of morning glory seeds is significantly reduced when the electric field intensity is 500 kv/m. A suitable high-voltage electrostatic field can improve seed vigor and make seedlings grow vigorously, thereby improving the ability of seed growth to absorb nutrients and transform and utilize energy substances.

#### 3.1.2. Food Thawing

As shown in Figure 4, the high-voltage electrostatic field thawing technology is a new type of non-thermal processing method. In the small area around the needle electrode, the electric field is used to accelerate the generation of ions. The generated momentum is transferred from the air ions to the neutral air molecules, and corona wind drives a large amount of fluid to the surface of the material, which in turn leads to an increase in the heat transfer coefficient [50,51]. Foods placed between the electrodes are processed by the applied electromagnetic field to achieve thaw. Therefore, the high-voltage electrostatic field thawing has the characteristics of high efficiency and energy saving. Compared with air thawing, high-voltage electrostatic field thawing can shorten the thawing time by half without affecting the thawing quality. Hsieh et al. used the electrostatic field to thaw tilapia meat [52]. They found that the high-voltage electrostatic field can effectively reduce the cooking loss of chicken by 1% to 3% and increase the water holding capacity by 1% to 2%. He et al. discovered the current under negative voltage is greater than the current under positive voltage for the same applied voltage. The corona wind speed changes within a certain range under a given applied voltage and electrode distance, and it is more stable under negative voltage than under positive voltage. Moreover, the current increases with the increase of the applied voltage, and since the corona wind speed increases with the square root of the current, the thawing rate increases under a high voltage and large current [53]. The rapid thawing characteristics of the high-voltage electrostatic field have also attracted the attention of scholars. Dalvi-Isfahan et al. investigated the voltages and freezing temperatures of thawing meat in a high-voltage electrostatic field [54]. The results demonstrated that a slight increase in the hardness value could be seen with an increased voltage or decreased freezing temperature. However, only changing the intensity of the voltage plays a significant role in the equivalent diameter, hardness, and drip loss. Li et al. used the negative electrons generated by the high-voltage electrostatic field to thaw the samples [55]. Defrosting under the −12 kv high-voltage electrostatic field treatment significantly reduced the water loss and microbial contamination of the fish pieces. In addition, it can enhance AMP-deaminase activity, reduce ACP activity, and delay IMP degradation. Therefore, the method can not only effectively maintain the freshness of the meat, but it also has a faster thawing speed. Rahbari et al. discovered that the fibrils of chicken breast stayed relatively intact without any gaps under 2.25 kv/cm. However, the largest deformation and loss of collagen fibrillation network appear under 3 kv/cm. The full results are shown in Figure 5 under different intensities voltages [56].

As shown in Table 2, the high-voltage electrostatic field has a significant effect on thawing meat products according to the related indicators. Some scholars have discussed the quality of food thawing on the structure and size parameters of the electrode plate. Amiri et al. determined that increasing the number of electrode needles can improve the oxidation degree of beef by fixing the electrode plate spacing [57]. Mousakhani-Ganjeh et al. showed that the oxidation degree of tuna will increase as the distance between electrode plates decreases [58]. Jia et al. found that there is no significant difference between high-voltage electrostatic field thawing and air thawing [59]. Consequently, the oxidation problem caused by the high-voltage electrostatic field can be solved by selecting the appropriate voltage intensity, plate spacing, or the number of electrode plates.

#### 3.1.3. Food Preservation

The mechanism of electrostatic field preservation is quite different from that of plant growth. The electrostatic field plays a positive role in the germination of seeds and plants to a certain extent, which strengthens the metabolic process [60]. However, preservation is to delay the aging process of fruits and vegetables by inhibiting the respiratory metabolism of fruits and vegetables. There is still a big controversy in the preservation mechanism. Some researchers believe that the electric field affects the metabolic activity of biological cell membranes by changing the transmembrane potential. Others hold the view that the biological electric field inside the fruits and vegetables affects the electron transporter in the respiratory system.

Zhao et al. used a high-voltage electrostatic field to treat green mature tomatoes for 2 h at 20 °C. The high electric field apparatus contains a power generator, treatment chamber, two stainless steel plates, one high-voltage controller, one voltmeter, and one amperometer. It can output −30–0 kv and 0–30 kv maximum, respectively. The results shown that the respiration rate of vegetables after electric field treatment decreased, which prolonged its storage time [61]. Kao et al. treated fresh-cut broccoli with 50–400 kv/m high-voltage electric field at 4 °C. The results show that the high-voltage electric field can prolong the storage time of fresh-cut broccoli according to the analysis of color and vitamins [62]. Takaki et al. introduced vegetables and fruits with a high-voltage electrostatic field. Compared with the control group, the storage time of the juice after the treatment was prolonged [63]. As for meat products, Wen-Ching Koa et al. used 100 kv/m high-voltage electrostatic treatment on tilapia. The equipment included an electrostatic generator, whose output voltage was 50 kv. This technology can delay the water and salt soluble to prolong the fresh-keeping period of fish by ~2 days [64]. Moreover, the 300 kv/m high-voltage electrostatic can inhibit actomyosin Ca^2+^-ATPase activity. If a 600 kv/m high-voltage electrostatic is applied, a longer extension time can be obtained.

### 3.2. Extremely Low Frequency Electromagnetic Field (0–300 Hz)

The extremely low frequency electromagnetic field (ELF-EMF) is the most common electromagnetic field because the power frequency of most countries in the world is in this frequency band. ELF-EMF has a certain effect on humans, including immune function, cell differentiation, cell proliferation, and transmission of intracellular signal substances. It can achieve cell differentiation, tissue repair, differentiation of stem cells, and healing of injured cells. Specific ELF-EMF can stimulate the production of local bone factors, and significantly improve the proliferation and differentiation indicators of bone density. In addition, ELF-EMF can also inhibit tumor growth and induce cell apoptosis [65,66]. More important, the human nervous system is the most sensitive issue to ELF-EMF. A large number of experiments have shown that the effect of ELF-EMF on the central nervous system is double-sided. On the one hand, the ELF-EMF may affect the normal brain development. It has an important impact on the growth and differentiation of neurons and the direction of neurites. On the other hand, electromagnetic fields may cause oxidative stress in the brain tissue, which leads to the activation of astrocytes. Based on the above research, this review focuses on the application of extremely low frequency electromagnetic sensors in cancer treatment, fracture healing, peripheral nerve repair, and transcranial magnetic stimulation.

#### 3.2.1. Cancer Treatment

ELF-EMF can induce apoptosis of cancer cells to some extent, thus providing a new way for the treatment of cancer. Studies have shown that ELF-EMF have significant therapeutic effects on a variety of tumors. They can induce growth inhibition and tumor killing effects on lung cancer, liver cancer, breast cancer, prostate cancer, and melanoma cells [67,68,69]. The main anti-tumor mechanisms of electromagnetic fields can be roughly divided into the following four categories:Induce the level of ROS in tumor cells to increase, which in turn damages DNA, proteins, and membrane lipids;Produce selective cytotoxicity to tumor cells and achieve an anti-tumor effect by improving cellular immunity;Directly damage the DNA chain to cause chromosome aberrations and inhibit tumor growth;Promote tumor cell apoptosis and cell cycle arrest.

The rotating electromagnetic sensor has a significant effect on tumor suppression. In 2011, Wang et al. found that the magnetic field with an intensity of 0.4 T and excitation frequency of 7.5 Hz for 4 days can inhibit the proliferation of a variety of human tumor cells, including gastric cancer BGC-823, gastric cancer MKN-28 cells, and lung cancer A549 [70]. However, this technology does not inhibit poorly differentiated gastric cancer MKN-45 cells and lung adenocarcinoma SPC-A1 cells. It cannot inhibit poorly differentiated gastric cancer MKN-45 cells and lung adenocarcinoma SPC-A1 cells. Ren et al. found that the rotating magnetic field with an intensity of 0.4 T and excitation frequency of 7.5 Hz for 6 days can induce the senescence of lung cancer cell A549 and inhibits its iron metabolism [71]. If the magnetic field intensity and frequency are changed, for example, the rotating magnetic field with an intensity of 0.4 T and excitation frequency of 7.5 Hz for 15 min does not have a significant effect on the proliferation of HepG2 liver cancer cells. The research shows that the intensity of the magnetic field, the processing time and the cell type are all key factors for the effect of magnetic field on cells.

Zhang et al. designed a gyromagnetic sensor composed of two groups of 30 neodymium iron boron permanent magnets which are anti-parallelly stacked. The structure of sensor presented in Figure 6a [72]. The magnet array rotates on the vertical axis at a frequency of 8–10 Hz. The MF intensity at the top of the processing table is 0.6 T, and the MF intensity at 5 cm above the surface is 0.4 T. Moreover, the horizontal spatial change in the animal cage region is 0.6–0.32 T. The rotation introduces an approximately sinusoidal 0.21 T average fluctuation in the MF intensity with a frequency of 8–10 Hz. As shown in Figure 6b, Du et al. investigated the gyromagnetic sensor with a cylindrical permanent magnet rotating around the long axis [73]. Some scholars studied the gyromagnetic sensor composed of NdFeB Halbach array magnets. As shown in Figure 6c,d, a uniform 1 T magnetic field is generated radially in the center part of the device, and a magnet is installed on the motor to control its rotation [74]. Under a rotating magnetic field, the internalized MPs can effectively destroy the cell membrane through mechanical force and promote programmed cell death in vitro and in vivo. Magnetic field therapy successfully reduced the size of brain tumors and prolonged survival rates. The development of the above-mentioned sensors can significantly improve the uniformity of the magnetic field and accelerate the induction of apoptosis of some cancer cells to a certain extent.

Sisken et al. used pulsed electromagnetic fields with an intensity of 0.3 mT and a frequency of 2 Hz to treat rat sciatic nerve defects, which proves that electromagnetic fields can stimulate the regeneration of the nervous system [75]. Experimental results show that the application of pulsed electromagnetic fields can accelerate the formation of myelin and connective tissue [76]. Zhong et al. measured the proliferation and differentiation of bone marrow mesenchymal stem cells under condition of the electromagnetic field with an intensity of 0.5 mT and frequency of 50 Hz [77]. The results confirmed that prolonging the stimulation time can promote the cell proliferation and induce the cell differentiation. After transplantation into animals, the original cells of bone marrow mesenchymal stem cells stimulated by the magnetic field performed obvious biological characteristics. Diniz et al. used a Helmholtz copper coil to analyze the effect of pulsed magnetic fields on the proliferation of osteoblasts, which also confirmed the effectiveness of electromagnetic stimulation. The diameter and number of turns are set to 12 cm and 250, respectively. The excitation frequency of the coil is set to 15 Hz [78]. Liu et al. designed a set of ELF-EMF biosensors consisting of a driving unit and a pair of Helmholtz coils. The diameter of the coil is 14 cm, and the distance between the coils is 13 cm. The induced peak voltage and the change intensity of the magnetic field are ±0.7 mV and 0.4 T/s, respectively [79]. They found that the pulsed magnetic field generated by the electromagnetic sensor can protect the synthesis of chicken sternum cartilage proteoglycan in the embryo.

#### 3.2.2. Transcranial Magnetic Stimulation (TMS)

TMS is based on the principle of electromagnetic induction, which applies a rapidly changing current to the coil to generate a strong and short-lived magnetic field, as shown in Figure 7 [80]. When the voltage of the induced electric field reaches a certain strength, the resting membrane potential of the cell will change. Thereby, it has an excitatory or inhibitory effect on the nervous tissue to affect the physiological and biochemical activities of the brain, and finally achieves the therapeutic effect on neurological and mental diseases.

In the application of cranial nerve stimulation, the magnetic field generates an induced electric field in the brain tissue to form an intracranial current, thereby changing the stimulation location and related neuronal activity. In 1985, Anthony Baker et al. were the first to study TMS. Nowadays, TMS has been widely used in clinical experiments and research in the fields of neurology, neuroscience, psychiatry, rehabilitation, psychocognitive science, and sports medicine [81]. It has a good therapeutic effect in depression, obsessive-compulsive disorder, schizophrenia, and peripheral nerve rehabilitation [82].

Jiang et al. used TMS to treat 40 patients with chronic primary insomnia. The results show that TMS treatment significantly improved the sleep cycle of stage III sleep and rapid eye movement sleep [83]. Levitt et al. used 10 Hz TMS to treat patients with refractory depression [84]. Magnetic resonance spectroscopy is used to assess the changes in GABA levels of the left dorsolateral prefrontal cortex (DLPFC). The results indicate that TMS treatment is associated with elevated GABA levels, and subjects who receive GABA agonist treatment have less response to TMS. Leblhuber et al. studied the changes in the utilization of neurotransmitter precursor amino acids in elderly patients with depression after stimulation of the prefrontal cortex, and found that TMS has a significant effect on the treatment of depression [85].

The performance of the TMS system is mainly determined by the focus, stimulation depth, and intensity of the induced electric field. In fact, the structure of the TMS coil is directly related to the electric field distribution in the brain tissue. Among them, the depth and focus of the induced electric field distribution are the two most critical indicators for the optimal design of the TMS coil [86]. In order to locate the optimal stimulation point of the sensor to the intracranial cerebral cortex, the structure of the sensor needs to be precisely designed. It is helpful to make continuous breakthroughs in the performance of TMS by simulating the performance of the coil in terms of stimulus intensity, stimulus depth and focus.

(1)Circular coil

The structure of the circular coil is simple, and the induced electric field on the surface of the brain is distributed in a “circular shape.” The stimulation performance of the circular coil is poor, especially in focus behavior. In order to improve the focus, one side of the coil can be folded up along the center of the circle. The novel structure of coil is called the deformed circular coil. The simulation results show that the magnetic field induced by the deformed circular coil has a large attenuation. The change in the magnetic field leads to a change in the electric field distribution, which in turn results in an improved focusing. Figure 8a shows the structure of a circular coil and a deformed circular coil.

(2)Figure-of-Eight coil

Figure-of-eight coils are widely applied in commercial applications. Two coils with opposite currents cause the peak of the induced electric field to appear at the tangent position of the coil [87,88]. Figure 8b shows the figure-of-eight coil. Compared with the circular coil, the focus of the figure-of-eight coil is greatly improved. However, it can only be applied to the stimulation of the superficial areas of the brain. In order to further improve the performance of the figure-of-eight coil, a butterfly coil and a quadruple butterfly coil are proposed. The structure of the butterfly coil and quadruple butterfly coil are improved based on the figure-of-eight coil. By enhancing the magnetic field at the tangent of the coil, the focusing performance is improved and the target point has greater stimulation intensity. In addition, the quadruple butterfly coil can be used in combination with the shielding plate, which can further reduce the focal area of the brain surface.

(3)H-shaped coil

The H-shaped coil does not cause high electric field strength in non-target areas because it has a small electric field attenuation rate and surface charge accumulation. As shown in Figure 8c, it can effectively stimulate the deep neuron area, so it belongs to the deep stimulation coil. In addition, the penetration depth of each area can be flexibly controlled by adjusting the coil input or changing the distance between the coil and the brain [89,90].

(4)Halo coil

A large diameter circular coil placed in the middle of the head is called the Halo coil. The special position of the Halo coil enables it to stimulate deeper tissues in the brain; thus, the Halo coil belongs to the deep stimulation coil category. Figure 9 and Table 3 show the structure and model characteristics of the Halo coil, respectively. Halo coils can also be combined with other coils, such as the Halo-circular assembly coil (HCA coil), Halo coil working with two circular coils (HTC coil), Halo and Fo8 coils (HFA coil), double-cone coil with the Halo coil (HAD coil), and the triple-halo coil (THC coil) [90,91,92]. Although different coils have different structures and characteristics, they all realize the superposition and cancellation of the magnetic field by changing the direction of the input current, which increases or reduces the intensity of the induced electric field.

Moreover, there are linear coil arrays, cap-formed coil arrays, multi-layer coil arrays, etc. [93,94]. The structures of the coil arrays are shown in Figure 10. The linear coil array has a small number of input control units, and the stimulation mode is flexible and changeable by changing the size and direction of the input current. The cap-formed coil array has a greater stimulation depth and weakens the impact on non-target areas due to the increased coupling with the head surface. The multi-layer coil array increases the stimulation intensity to a certain extent. As the number of coils increases, the number of channels of the magnetic field generator should also increase accordingly. In other words, the overall structure and working conditions are more complicated, and hence, the multi-layer coil array has poor applicability.

### 3.3. Intermediate Frequency Electromagnetic Field (300 Hz–10 MHz)

Intermediate frequency electromagnetic detection are widely applied in areas such as food quality and spatial location monitoring. In food quality monitoring, the organization, composition, structure and state of meat products, fruits, and vegetables are closely related to their electromagnetic properties [95]. For example, the impedance value has a certain change rule in the process of meat decay. In addition, the impedance value of the electromagnetic sensor is also closely related to its position, so the electromagnetic sensor can detect changes in the relative position in space. At present, this technology is mainly used for the monitoring of human joints, limbs, and eye movements. The applications of electromagnetic sensors in food quality and spatial location monitoring are introduced as follows.

#### 3.3.1. Food Quality Inspection

The decay of meat products produces many complex biochemical and physical and chemical processes, which will affect its organization and structural characteristics [96]. These phenomena also change the electrical parameters. Biological tissue can be considered as a non-magnetic material, and its electrical properties are the result of the joint action of its tissue and structure. Among them, the most important component that affects the dielectric properties of the tissue is the ion. As for the structure of the dipole moment, the main factors are water tissue molecules, proteins, and lipids. The movement of electric charges causes conduction effects, and the polarization of dipoles produces dielectric relaxation. The conductivity of most tissues is low when electromagnetic waves are excited at low frequencies, which mainly depends on the volume fraction of extracellular fluid. As the excitation frequency increases, the conductivity of intracellular and extracellular ions rises sharply due to the dielectric relaxation of water. The whole process can be represented by Figure 11.

The above-mentioned increase in conductivity is called dispersion. In biological systems, there are mainly three kinds of dispersions (*α*, *β*, *γ*). According to the phenomenon of dispersion, it is divided into three frequency bands. The frequency range of α dispersion is from millihertz to hundreds of hertz, and it is mainly related to the phenomenon of tissue polarization. The frequency range of *β* dispersion caused by the Maxwell-Wagner effect is from hundreds of hertz to megahertz. It is a method of measuring the integrity of cell membranes during meat aging because of reduced insulation properties. The frequency range of the *γ*-dispersion is in gigahertz, which is mainly caused by the permanent dipole relaxation of free water in muscle tissue. This characteristic also further proves the reliability of the electrical method for detecting meat products. In fact, electrical detection can be divided into contact measurement methods and non-contact measurement methods. The response speed of the electrode contact measurement method is fast and accurate. However, the contact method may cause unnecessary measurement errors, such as the chemical reaction and food tissue damage. This problem does not exist in non-contact measurements. As a non-contact measurement method, the electromagnetic detection method has a wide range of applications in food quality detection [97].

Swatland et al. reported the changes in rheological properties, electrical properties, and optical properties during the ossification of pig and cattle carcasses [98]. This process can be detected by its low reactance and low capacitance, and it is found that these three physical detection characteristics have similar variation curves. Ibba et al. studied the influence of frequency on the bio-impedance of fruits using electrical impedance spectroscopy. They pointed out that the bio-impedance can be used to measure fruit quality. The results demonstrated the potential of this technique for monitoring the ripening of fruit, obtaining a good correlation of the impedance evolution with low frequency points [99]. Sun et al. combined chemometrics and impedance characteristics to create a multi-mass index for Atlantic salmon. Moreover, they applied a partial least squares method to create a quantitative prediction model of the bioimpedance spectroscopy and total volatile basic nitrogen value. The correlation coefficients of the training set and test set are 0.9447 and 0.9387, respectively, which means that the impedance characteristics measurement method has the advantages of convenience, economy, and high precision. Castro-Giraldez et al. noted that the certain frequencies (500 Hz, 300 kHz) are effective at 24 h after slaughtering to detect the dark firm dry (DFD) meats during the post-mortem period. In fact, there may be systematic errors in the measurement [100]. Kim positioned the monopolar injection needle to the dermis, subcutaneous tissue, or muscle layer of pork tissue, and then measured electrical impedance in the frequency range of 10 Hz to 10 kHz based on an impedance analyzer [101]. This method has also been proven to be used for pork quality monitoring. Tang et al. introduced a non-contact induction measurement system for bioimpedance spectroscopy [102,103,104]. The electrical conductivity of the food is measured by an excitation-retraction electromagnetic sensor in the frequency sweeping mode, and the freshness of the food is detected according to the change of the electrical conductivity signal. Apart from that, Tang et al. used the fast finite element method to establish the cell model, as shown in Figure 12. The solution of the numerical model can obtain the vortex field distribution inside the cell and further proves the accuracy of the measurement results.

#### 3.3.2. Spatial Position Measurement

Electromagnetic sensors can be used for spatial position measurement. When a small receiving sensor moves in space, the electromagnetic positioning system can accurately calculate its position and orientation, thereby providing dynamic and real-time measurement of the position and orientation angle [105,106]. In biomedical research, it is an ideal choice for measuring range of motion and limb rotation due to the advantages of being fast and precise. The electromagnetic tracking system generally consists of a magnetic field transmitter and receiver. A magnetic field generator is placed near the target, and the magnetic field covers a certain range. The receiver detects the strength and phase of the magnetic field and sends the signal to the control unit. The control unit can calculate multiple degrees of freedom for tracking the target. The advantage of the electromagnetic tracking system is that the detection result is not restricted by the line of sight. Nothing can block the tracking of the electromagnetic tracking system except for electrical conductors or permeable magnets. The disadvantage is that it is easily interfered by metal and the working range is small.

Recently, a three-dimensional electromagnetic tracking system was launched, as shown in Figure 13. It can measure the range of motion of the lumbar spine, cervical spine, and shoulders of humans. The “Flock of Birds” electromagnetic tracking device is used for three-dimensional motion analysis, and the patient’s arm is extended in three directions [107]. The slope of the regression line between the rotation of the scapula joint and the rotation of the scapula is calculated, which reflects the rhythm of the scapula. According to the measurement result, the movement information of the shoulder joint can be obtained in time.

In accordance with the idea, Xie et al. proposed a new method of biomechanical motion monitoring based on electromagnetic sensing technology [108]. The proposed method has the advantages of non-invasiveness, easy implementation, low cost, and high efficiency. A theoretical model was established to model the sensor response, and the wearable sensor system and corresponding measurement signal shown in Figure 14 was designed to monitor various biomechanical movements, including blinking frequency, finger/wrist bending level, and frequency. The whole system is measured by the mutual impedance of the coil. According to the change frequency of the signal per unit time, it can reflect the body’s movement state in time. Therefore, this technology can be used to detect early signs of these abnormal biomechanical behaviors, so that relevant treatment can be carried out in time. Robison invented an electromagnetic method of eye movement recording, which has the characteristic of low noise; however, the method is limited by blinking and fluctuations in pupil size [109]. Remmel designed a cheaper version. The system uses three alternating magnetic fields (48, 60, and 80 kHz) in the X, Y, and Z directions to record eye movement information [110]. The subject wears a contact lens with a small loop of wire, and his eyes are located close to the center of the test field. The three alternating magnetic fields introduce three voltages into the coil through Faraday’s law of induction. As the eye rotation, more magnetic field passes through the coil, which generates induces larger voltage. Three analog (direct current) voltages of the vector components in the X, Y, and Z directions are obtained by amplifying and demodulating the coil signal. According to the measurement signal, the angle information of the eye can be inverted. Later, Remmel extended this method to arm recording [111]. The two small coils fixed at the elbow and two coils fixed at the wrist are applied to measure the movement of the arm. For each coil in the system, the voltage is demodulated to give three DC voltages to obtain the vector direction of the coil axis. At the same time, the inhomogeneity of the magnetic field is corrected. This method can describe five angles of arm movement, and the noise is only 2 arcsec rms.

### 3.4. Radio Frequency Electromagnetic Field (10 MHz–300 GHz)

Radio frequency electromagnetic fields (RF-EMFs) generated by communication base stations, mobile phones, wireless phones, and wireless routers have become an indispensable part of modern life. The non-thermal effects of radio frequency electromagnetic fields have a serious impact on daily lives [112]. Although the non-thermal effects of radio frequency electromagnetic fields have a serious impact on people’s daily life, electromagnetic fields can react with biological tissues for disease detection. Therefore, this review will introduce some applications and developments of high-frequency electromagnetic detection in disease detection.

#### 3.4.1. Disease Detection Based on MRI

Nuclear magnetic resonance (NMR) technology is mainly based on the interaction between magnetic nuclei and magnetic fields. The nuclear magnetic moment of the low-energy state absorbs the energy provided by the alternating field strength, and then continuously transforms to the high-energy state, thereby generating nuclear magnetic resonance signals [113]. This transition phenomenon caused by resonance absorption is called the nuclear magnetic resonance phenomenon.

MRI is based on the nuclear magnetic resonance signal emitted by the hydrogen nuclei in the human body. The strength of this signal depends on the position of the hydrogen nucleus in the molecular structure and the surrounding environment, so it can reflect a large amount of media information. The strength of this signal depends on the position of the hydrogen nucleus in the molecular structure and the surrounding environment, so it can reflect a large amount of media information. In the late 1980s, the phased array coil concept was introduced into the field of magnetic resonance imaging, which revolutionized the reception of MR signals [114]. The receiving array can use multiple small surface RF coils to obtain a higher signal-to-noise ratio image in a wider range by appropriately combining the signals from a single element. Since the efficiency of parallel imaging and the achievable acceleration factor depend on the number of coils in the receiving array, the number of receiving channels in the MR system has been increasing over time [115]. In fact, most research is based on phased array coil technology to reduce the coil size and increase the number of channels, which can improve the imaging quality [116]. Compared with traditional mechanical coils, wearable technology can maximize the fill factor of the magnetic resonance receiving coil to significantly improve the imaging quality [117].

The Institute of Biomedical Engineering of the University of Zurich in Switzerland has made various attempts to implement wearable technology [118,119]. As shown in Figure 15, the fabricated coil adopts an adjustable mechanical structure to adapt to the size change of the measured part. This kind of coil has a complex structure and large volume, which determines that it is difficult to have a wide range of applications. Subsequently, the team used thin wires to achieve the softness and scalability in a single direction by braiding a wire harness. In in vivo knee imaging, it was demonstrated that a stretchable array of eight receiver coils allows adaptation to different anatomical dimensions and different knee flexion angles.

In order to explore the relationship between the number of phased array coils and the ultimate performance, the GE global research team created a 128-channel “flexible” coil for torso imaging. As shown in Figure 16a, this coil can change the shape of the coil within a certain range. However, the huge geometric size and limited deformation space are not suitable for infants and weak patients [117]. The signal-to-noise ratio (SNR) in the proposed system is 1.03 in the center of the elliptical loading model and 1.7 in the outer region on average. Furthermore, the maximum g factor of 4 × 4 acceleration is reduced to 2.0, and the 5 × 5 acceleration is reduced to 3.3. These characteristics significantly reduce the residual aliasing artifacts in human imaging.

As shown in Figure 16b, Jia et al. deployed coils on light and soft materials, but the coil array cannot be stretched. The feature of extremely flexible makes it suitable for inspections in many scenes [120]. The experiments show that the proposed MRI system can achieve a high (SNR) of 336 in vitro, and the phantom images are evenly distributed. As for in vivo experiments, it has a higher SNR of 169 in the region of interest, and the longitudinal coverage exceeds 48.5 cm. However, the bulky system line is still the biggest obstacle affecting its application. As shown in Figure 16c, Corea et al. used inkjet printing and screen-printing techniques to make coils [121]. At the same time, the relationship between the printing coil inductance and the “distributed capacitance” in different printing materials was analyzed. The results shown that the relative dielectric constant increases linearly with the concentration of barium titanate in the ink. In addition, the fabricated coil can be attached to the tested tissue, which is beneficial to improve SNR. The cost of this technology is relatively high, and the coils made by printed materials are less flexible. Therefore, materials that can meet the printing technology and high flexibility are still the bottleneck of the realization of this magnetic resonance coil.

#### 3.4.2. Disease Detection Based on Microwave

The main task of microwave detection is to simulate the scattering and diffraction of the electromagnetic field passing through the medium [122]. This technology measures the electromagnetic field scattering information of the object medium to reconstruct the dielectric properties and geometric characteristics of the object. Since the quality of the microwave image is greatly affected by the dielectric properties of the tested target, the existing microwave imaging methods are mainly used for breast cancer detection. Only a few studies are focused on the detection of strokes.

In breast cancer detection, Wang introduced a fast microwave sensor for breast cancer detection through near-field imaging methods [123]. The method of detecting cancer cells based on microwave imaging is highly correlated with the contrast of dielectric properties between healthy tissue and malignant tissue.

Bassi et al. investigated the confocal microwave imaging (CMI) for breast tumor detection. The results show that the 2D CMI system can detect small tumors (diameter 2 mm), and the 3D CMI system can identify tumors of medium size (diameter greater than 6 mm). Moreover, they applied delayed multiplication and summation methods in CMI to reduce artifacts and noise and enhance the image [124]. Yang et al. first proposed the detection method of the Microwave Image for breast cancer detection based on the principle of ground penetrating radar [125]. It uses the transmitting antenna to emit Ultra-Wideband (UWB) microwaves to irradiate the breast, and then the receiving antenna to receive the scattered signal. According to the scattered signal, the strong scattering area in the breast is recovered to determine the position of the tumor. Rappaport et al. used the time reversal algorithm of Finite-Difference Time-Domain (FDTD) to realize the spatial reconstruction of the dielectric constant distribution inside the breast. They deduced the iterative formula of FDTD to obtain the initial distribution of the electromagnetic field in space [126,127]. The method is an ill-conditioned solution problem, for which huge challenges exist in robustness and computational cost.

As for brain stroke detection, researchers at the University of Queensland in Australia have designed an ultra-wideband microwave system for head imaging [128]. According to the construction of the human brain model, the confocal algorithm is used to realize the detection of stroke. Ireland et al. used the Born iterative method to detect hemorrhagic stroke in brain tissue [129]. Furthermore, a high-contrast brain conductivity distribution map is reconstructed through microwave imaging technology. The internal information can be detected according to the distribution of brain conductivity. The reconstruction results of conductivity can be used to detect strokes.

## 4. Application of Machine Learning Technology in Electromagnetic Medical Images

Artificial intelligence is a technological science which can be used to simulate and expand the theories and methods of human intelligence. Machine learning (ML) is a method of realizing artificial intelligence, which applies algorithms to analyze data, make decisions, and predict specific events. Deep learning (DL) is a technology that realizes machine learning. It is a data-driven method of automatically learning high-level features hidden in images, which can greatly reduce the interference of subjective factors in feature selection [130]. In addition, the model applies a non-linear layer structure, which can be used to build a more complex model. Shallow neural networks are prone to under-fitting, while DL can improve the learning ability by increasing the depth of the network to solve more complex problems.

According to the electromagnetic biological effect, the signal detected by the electromagnetic biosensor can reflect the information of the biological body. However, manual analysis and interpretation are still required. In fact, machine learning can be used to learn the characteristics of biological signals, instead of manual operation. In particular, machine learning algorithms have been widely used in medical image diagnosis. Therefore, this review summarizes the application of machine learning in microwave breast cancer images, MRI prostate cancer images, and MRI brain tumor segmentation images.

### 4.1. Application of Machine Learning in Microwave Breast Cancer Images

In recent years, researchers have begun to use ML methods to directly process the signals obtained by the microwave breast cancer detection system, and thus, achieve the purpose of breast cancer screening [131]. The method of machine learning is mainly divided into two research directions. The first one is to assume that the tumor has been confirmed to exist in the breast. According to the trained model, an evaluation of the physical characteristics is needed, such as the location and size of the tumor. The other is to use the model to determine whether the tumor exists.

Under the assumption that the tumor exists, Davis et al. used a combination of Local discriminant bases (LDB) and Principal Component Analysis (PCA) to estimate the morphological characteristics of the tumor, such as the shape and size of the tumor [132]. The detection results are shown in Figure 17. Under the condition of a 10 dB signal-to-noise ratio, the method can reach an accuracy of 97% for breast cancer detection, and the shape accuracy is 70%. Conceicao et al. took the shape of the tumor as a feature [133]. The features combined with support vector machine (SVM) are used to determine whether it is a malignant tumor. They first constructed a breast prosthesis and tumors of different shapes, then placed tumors of different shapes on the breast prosthesis for measurement to obtain training samples. After that, the PCA was used to reduce the dimensionality and feature extraction of the acquired data. Finally, they analyzed the data by an SVM classifier to establish the model.

On the assumption that the tumor cannot be determined to exist, Alshehri et al. first used Discrete Cosine Transform (DCT) to extract the characteristics of the received signal of the microwave antenna [134]. Then, an artificial neural network was used as a model for judging whether the tumor was presented. They verified the effectiveness of the algorithm in uniform and uneven breast implants. The results show that the tumor existence, size, and location detection rates for both cases are highly satisfactory, which are approximately: (i) 100%, 95.8%, and 94.3% for homogeneous cases and (ii) 100%, 93.4%, and 93.1% for heterogeneous cases, respectively. Byrne et al. used PCA to extract features of the received signal on the digital breast model. They selected the SVM as classifier on the signal of each channel [135]. The final decision result was voted by the classifiers of all channels according to the majority principle. The investigated method can detect tumors as small as 4 mm in diameter with an accuracy over 71% in a dielectrically heterogeneous breast. In 2014, Santorelli et al. used SVM linear discriminant analysis to classify breast cancer tumors, and achieved an accuracy of 76.71% for identifying signals from the tumorous phantoms [136]. The above studies have confirmed that ML technology can efficiently assist doctors in disease diagnosis. For one thing, it can reduce the burden on doctors; for another, it can reduce the subjectivity of diagnosis, thereby making the diagnosis more objective and accurate.

### 4.2. Application of Machine Learning in MRI Prostate Cancer Images

Prostate cancer (PCa) is the cancer with the highest incidence among men. It ranks as the second-leading cause of cancer death in men. PCa has an insidious onset, but the disease progresses rapidly. The tumor cells are prone to local infiltration or distant metastasis. Therefore, early diagnosis and timely treatment have more positive meanings for PCa patients [137]. MRI is a mature and non-invasive imaging technique. Relying on the good display ability of prostate tissue structure and surrounding tissues, it occupies an important position in the diagnosis of PCa. Machine learning technology is also used in prostate cancer MRI image monitoring, mainly including prostate segmentation and cancer detection.

The segmentation of the prostate is usually a necessary step for clinical setting and further image analysis. In order to save time and increase reproducibility, it is very meaningful to realize automated prostate contour and lesion segmentation. Perfect automatic segmentation can get a fully automatic post-processing pipeline. Yan et al. proposed a model based on deep propagation neuron network by extracting the data from different levels of complexity, which can obtain more credible segmentation results of glands and their boundaries as shown in Figure 18 [138]. In addition, Alkadi et al. applied a system based on unimodal deep learning. The system can automatically segment prostate and PCa lesions [139]. Their algorithm has high segmentation accuracy and can obtain an area under the curve (AUC) of 0.995.

In terms of prostate cancer detection, ML-based computer-aided diagnosis software has important applications in the PCa field. It can improve diagnosis accuracy and repeatability. McGarry et al. used ML to detect epithelial cells and luminal density [140]. The data of 10 patients remained stable, which means the method can be used to diagnose high-grade PCa. In addition, volume of index lesion ROI can evaluate histogram parameters obtained from T2WI, DWI, and DCE images. The combination of ROI and SVM can improve the PI-RAD Sv2 score.

### 4.3. Application of Machine Learning in MRI Brain Tumor Segmentation Image

As a typical non-invasive imaging technology, magnetic resonance imaging (MRI) can produce high-quality brain images without damage and skull artifacts. MRI can provide more comprehensive information for the diagnosis and treatment of brain tumors [141]. Doctors can quantitatively analyze brain tumors to measure the maximum diameter with the help of multi-modal brain imaging to segment tumors and determine the volume and number of brain lesions. MRI brain tumor segmentation has also become an important component in the field of medical imaging because of the needs of clinical application and scientific research [142]. Researchers have proposed many methods for the segmentation of brain tumors in MRI images, but the existing methods still cannot always achieve satisfactory results due to the complexity of MRI brain images. The deep neural network model has been successfully applied to many computer vision tasks, which has received extensive attention from the scholars [143]. In specific implementation, the deep learning method segmented a large number of small-scale image blocks from the original MRI brain image. Furthermore, the image blocks are used to train a classification network. During segmentation, the corresponding image blocks are classified pixel by pixel in the context of a sliding window, and then post-processing is used to complete the segmentation of the brain tumor. This method obtains a large number of re-labeled small image blocks from the limited labeled brain image samples. Therefore, it can solve the contradiction between the few brain tumor-labeled images and the large demand for deep neural network training samples to a certain extent. The basic framework of the model for MRI brain tumor segmentation can be described as in Figure 19. The whole network architecture consisted by convolution, activation, pooling, fully connected layers, and post-processing, including conditional random fields (CRF), long short-term memory (LSTM), etc. In view of the powerful ability of automatic extraction of high discriminative features, it is quickly applied to MRI brain tumor segmentation.

The structure of a single network is used for brain tumor segmentation. The input data are processed by convolution, pooling, and nonlinear layers to obtain the feature map. Subsequently, the fully connected layer and classification layer are used to predict the label of the corresponding category based on the extracted feature. Single-network brain tumor segmentation methods can be divided into 2D and 3D due to convolution dimensions. Table 4 lists the performance of some typical single-network MRI brain tumor segmentation methods on the BraTS dataset.

The 2D convolutional neural network has the advantages of occupying less resources and fast training for brain tumor segmentation. Some researchers have carried out 2D network brain tumor segmentation research. For example, Zikic et al. proposed a 2D brain tumor segmentation network by using the Alex-Net network structure [144]. Pereira et al. used a similar VGG-Net network structure to construct an automatic segmentation method for exploring the advantages of small 3 × 3 convolution kernels [35]. In addition, they explored that small convolution kernels can design deeper network architectures, which can effectively avoid overfitting by reducing the number of parameters. Dvorak and Menze proposed a local structure prediction method to improve the local segmentation ability of MRI brain images [145]. Randhawa et al. improved boundary classification of MRI brain images [146]. In order to design a more effective brain tumor segmentation model, Ben Naceur et al. adopted the idea of multi-network ensemble learning to achieve accurate brain tumor segmentation [147]. The segmentation results are shown in Figure 20. Cui et al. focused on dividing the image block into 32 × 32 or 13 × 13-pixel scales according to the different label, so as to achieve better segmentation accuracy [148]. This method improves the accuracy of brain tumor segmentation and lays the foundation for a series of later methods.

**Table 4 biosensors-11-00225-t004:** Evaluation results of MRI brain tumor segmentation based on single CNN network.

Reference	Dataset	Evaluation Index DSC		
		Intact Tumor	Core Tumor	Enhance Tumor
[141]	BraTS 2013	0.88	0.83	0.77
[35]	BraTS 2013	0.86	0.75	0.73
[145]	BraTS 2014	0.83	0.75	0.77
[146]	BraTS 2016	0.87	0.75	0.71
[147]	BraTS 2017	0.89	0.76	0.81

Compared with the 2D network model, the 3D network can capture more characteristic information. The related research results are shown in Table 5. Kamnitsas et al. and Casamitjana et al. used a combination of the 3D convolution kernel and dual-pass model to improve the ability of the network to capture 3D features in MRI brain images. Among them, Kamnitsas et al. constructed a dense CNN segmentation architecture based on 3D local blocks [149]. As shown in Figure 21, the method achieved a high segmentation accuracy for brain tumor image. The proposed network combines multi-scale information by using different input sizes for each path. Casamitjana et al. employed the same input size for the two paths so that different receiving areas can focus on different information [150]. For the purpose of further improving performance, advanced network, such as Resnet and DenseNet, are gradually being introduced. Meanwhile, 3D convolution and dual-path modes are gradually being adopted in combination. Kamnitsas et al., (2017) combined dual-path and residual connection blocks to construct a 3D dual-path residual segmentation network for brain tumor segmentation [151]. It can obtain better segmentation performance while using less training data and fewer filters. Chen et al. proposed a new structure that can hierarchically segment necrotic and non-enhanced tumors and edema around tumors [152]. Therefore, 3D-CNN can accurately classify each pixel in MRI brain images.

The generative adversarial network GAN was proposed by Google Brain researcher Goodfellow and others. It can be considered as the most representative unsupervised generative method in recent years [155]. As shown in Figure 22, the generative confrontation network consists of generator G, discriminator D, and real data X. The generator is used to generate approximately realistic objects, so that it is difficult to distinguish true from false. The discriminator is mainly used to identify the difference between the real object and the false object produced by the generator. In the training process, the generator and the discriminator overlap each other, and continuously optimize and produce the best results in the confrontation. GAN has the inherent advantages of finding the data distribution of the model by paying attention to the potential probability density of the data, which overcomes the shortcomings of relying on large amounts of data. As the most creative unsupervised generative network, the GAN has gained wide attention in many fields such as image classification and segmentation. In recent years, it has also been effectively migrated to medical image analysis tasks, including MRI brain tumor segmentation. Table 6 is the performance of the MRI brain tumor segmentation method based on the GAN.

At the end of 2016, the semantic segmentation model based on adversarial networks is proposed. The method treats the segmentation network as a generator to train the network [156]. Li et al. tried to use GAN for MRI brain image segmentation at the MICCAI conference [157]. They trained the generator network and the discriminator network together to reduce the difference between the synthetic label and the real label. Furthermore, the high-order loss terms are used to enhance the spatial continuity of the segmentation results. Rezaei et al. proposed a new end-to-end training conditional generation confrontation network CGAN based on the GAN network [158]. The segmentation network and the discriminator network are changed to U-Net and Markovian GAN, respectively. The two networks are simultaneously trained through backpropagation to improve the segmentation accuracy. In 2019, Rezaei et al. proposed a new confrontation network called voxel-GAN [159]. It is mainly used to alleviate the problem of data imbalance in the brain tumors segmentation, since the proportion of tumors or unhealthy areas is small. In addition, there are methods that also introduce a 3D conditional generation confrontation network to alleviate the problem of brain tumor data imbalance based on the weighted adversarial loss generated by network training.

**Table 6 biosensors-11-00225-t006:** Evaluation results of MRI brain tumor segmentation based on GAN network.

Reference	Dataset	Evaluation Index DSC		
		Intact Tumor	Core Tumor	Enhance Tumor
[143]	BraTS 2015	0.85	0.70	0.66
[157]	BraTS 2017	0.70	0.55	0.40
[158]	BraTS 2017	0.87	0.72	0.68
[159]	BraTS 2018	0.84	0.79	0.63

## 5. Future Research Directions

Currently, the stability and reproducibility of biological detection are still poor due to the shortcomings of instability and variability of biologically active units. Nowadays, biological detection will inevitably be greatly developed with the rapid development of biology, informatics, materials science, and microelectronics [160,161,162,163,164]. It is foreseeable that electromagnetic detection will have the following characteristics in the future:Integration and intelligentization

In the future, electromagnetic detection will realize information storage and memory, logical judgment, two-way communication, self-check, self-calibration, self-compensation, numerical processing, and other functions. The sensor measurement unit can become a standard unit, which can increase the portability and interchangeability of the detection system. In addition, electromagnetic detection can also be closely integrated with computers. Machine learning technology automatically analyzes the data collected by the sensor and provides analysis results more quickly and accurately, thus forming a full automation of analysis and detection.

2.Multi-sensor fusion technology

Electromagnetic detection technology can be combined with other detection technology, such as capacitive detection technology, ultrasonic detection technology, etc. The fusion data processing methods is applied to improve measurement performance. It can integrate electrical, magnetic, acoustic, thermal, force, and other multi-physical field measurements. Based on the redundant data between multiple sensors, the stability and robustness of the system can be enhanced, and the performance and stability of the system can be improved.

3.Portability

At present, flexible sensing technology has made considerable progress based on the development of soft materials, flexible electronics, and nanotechnology. The combination of electromagnetic sensors and flexible sensing technology is conducive to online and real-time monitoring. People can use portable electromagnetic sensors for daily autonomous measurement, making it possible for people to diagnose diseases and monitor food quality in daily life.

4.Mechanism Research

At present, there is no reasonable explanation for the mechanism of some electromagnetic biological effects. Mechanism research should focus on reflecting biological issues from the perspective of physics. The establishment of standardized experimental models is particularly important for mechanism research. At the same time, new theories and methods learned from the fields of mathematics, physics, chemistry, and life sciences are used to achieve a major breakthrough in the study of biological electromagnetic mechanism. Based on the electromagnetic biological effect mechanism, the application range of electromagnetic detection can be further expanded. More and more parameter information can be obtained according to the measured signal inversion. More important, the study of mechanism also contributes to the improvement of sensor sensitivity and stability.

5.Networked detection

With the rapid development of network communication technology, networked detection systems that link distributed sensor nodes and fusion centers through communication networks have gradually received widespread attention. The networked sensor system improves the flexibility and scalability of system design due to its advantages, such as less wiring, low cost, easy expansion, and maintenance. The biosensor integrates a wireless transmitter chip, which can automatically transmit the collected data to the server and display it on the mobile terminal in real time. People can check biological information anytime and anywhere to facilitate daily life. Therefore, sensor networking is also an important development direction with the application of the Internet.

## 6. Conclusions

Electromagnetic detection technology has developed rapidly during the 40-year development process, which has attracted an increasing number of scholars. Since bioelectromagnetic detection has different mechanisms at different frequency ranges, they are widely used in disease diagnosis and treatment, food monitoring, environmental testing, etc. In particularly, the development of MRI technology has pushed electromagnetic detection to its peak. Therefore, this review summarizes the electromagnetic biological effects and their mechanism of action. Considering the excitation frequency and intensity significantly affects the electromagnetic biological effects; this review takes the excitation frequency of the electromagnetic field as a clue to introduce the application of bioelectromagnetic detection. Moreover, electromagnetic detection in combination with ML technology has been applied in clinical diagnosis, since ML can extract the deep information of data. The research involving the application of ML technology to electromagnetic medical images is also discussed. Finally, we present an opinion on the future development trend of electromagnetic detection and make a conclusion on the reviewed biosensing techniques.

## Figures and Tables

**Figure 1 biosensors-11-00225-f001:**
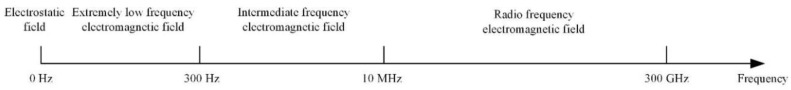
The frequency band of the electromagnetic field.

**Figure 2 biosensors-11-00225-f002:**
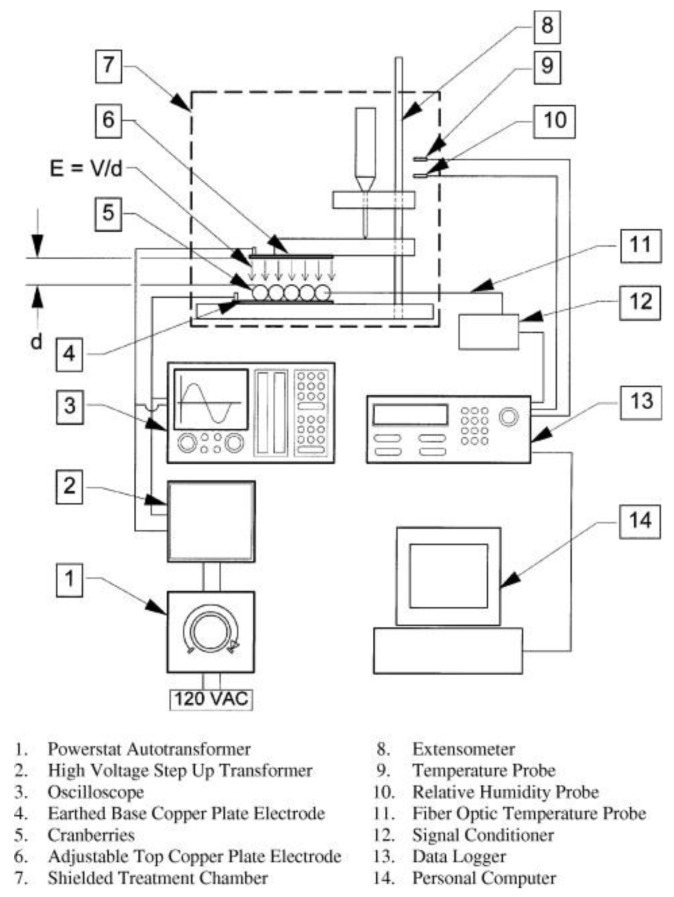
Schematic diagram of experimental set-up for high voltage field treatment of cranberry. Reprinted with permission from ref. [41]. Copyright 2008, Elsevier.

**Figure 3 biosensors-11-00225-f003:**
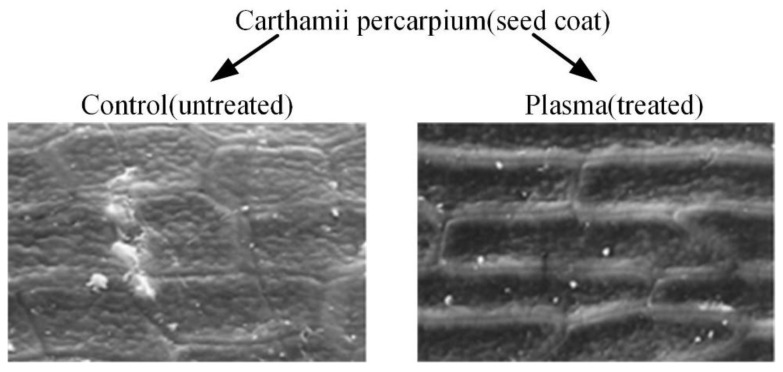
Seed germination processed by a high-voltage electrostatic field. Reprinted with permission from ref. [43]. Copyright 2006 Elsevier.

**Figure 4 biosensors-11-00225-f004:**
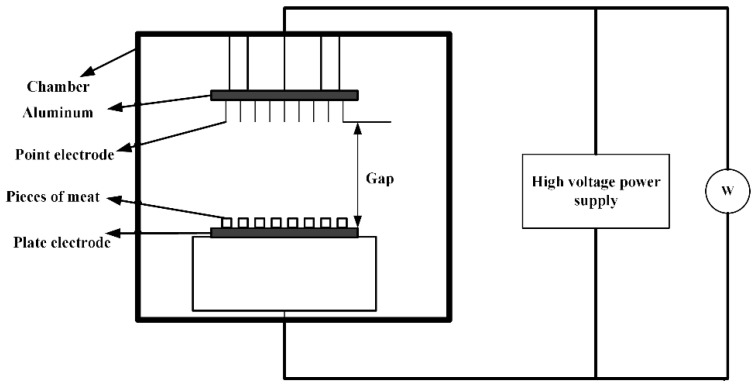
The schematic diagram of high-voltage electrostatic field thawing technology.

**Figure 5 biosensors-11-00225-f005:**
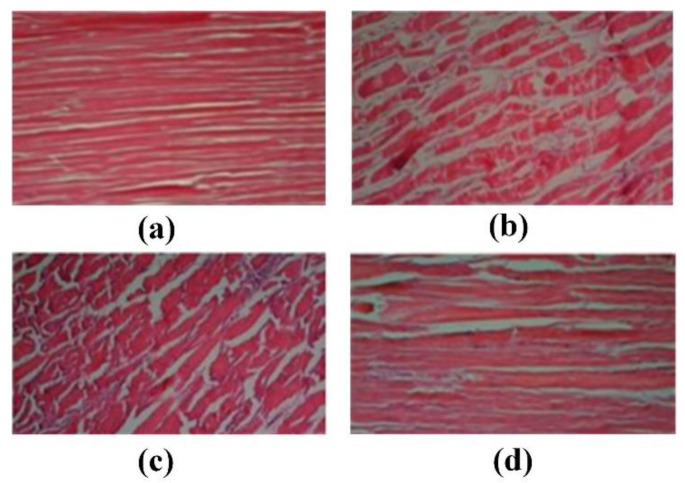
Sections of fresh chicken breast. (**a**) Thawed chicken breast by air method. (**b**–**d**) Thawed chicken breast by a high-voltage electrostatic field: (**b**) 1.5 kv/cm (**c**) 2.5 kv/cm (**d**) 3 kv/cm. Adapted with permission from ref. [56]. Copyright 2020 Wiley.

**Figure 6 biosensors-11-00225-f006:**
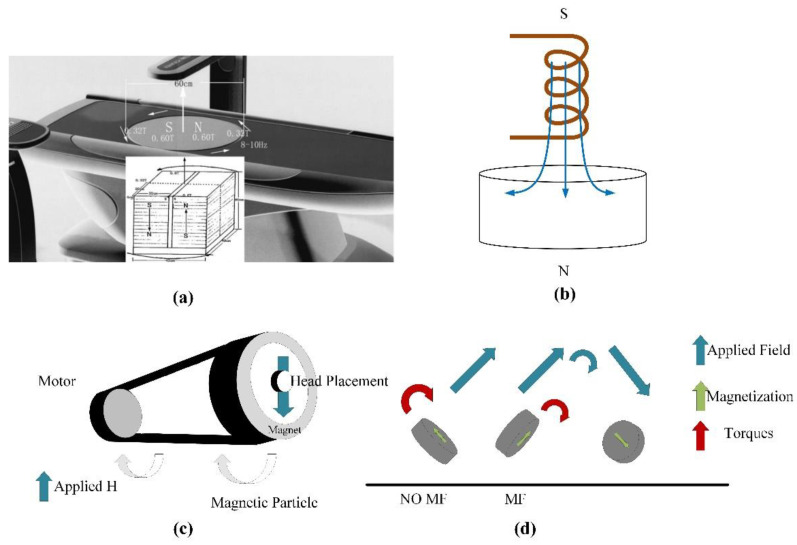
The structure of the gyromagnetic sensor. (**a**) Two anti-parallel stacks of 30 neodymium-iron-boron permanent magnets. Reprinted with permission from ref. [72]. Adapted with permission from ref. [102]. Copyright 2005 Wiley-Liss. (**b**) Cylindrical permanent magnet rotating around the long axis. (**c**) NdFeB Halbach array magnet. (**d**) The changes of the magnetic field corresponding to (**c**).

**Figure 7 biosensors-11-00225-f007:**
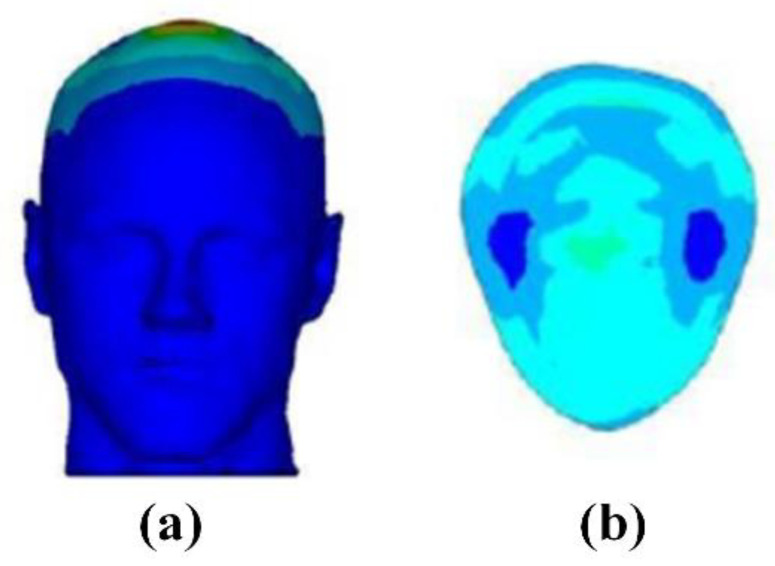
Magnetic field distribution stimulated by TMS. (**a**) Front view of the induced electric field distribution on the scalp surface. (**b**) Sectional view of the x-y plane.

**Figure 8 biosensors-11-00225-f008:**
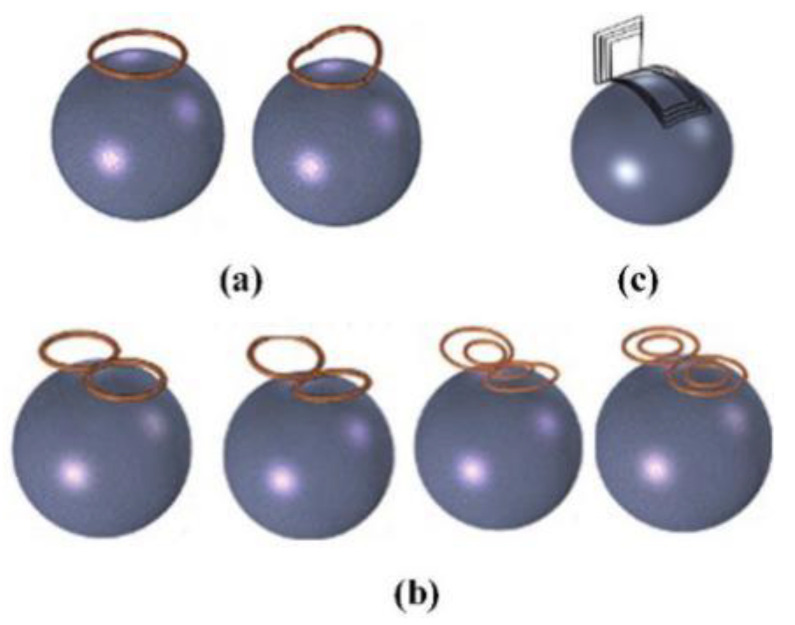
The structure of the TMS coil. (**a**) Circular coil. (**b**) Figure-of-eight coil. (**c**) H-shaped coil.

**Figure 9 biosensors-11-00225-f009:**
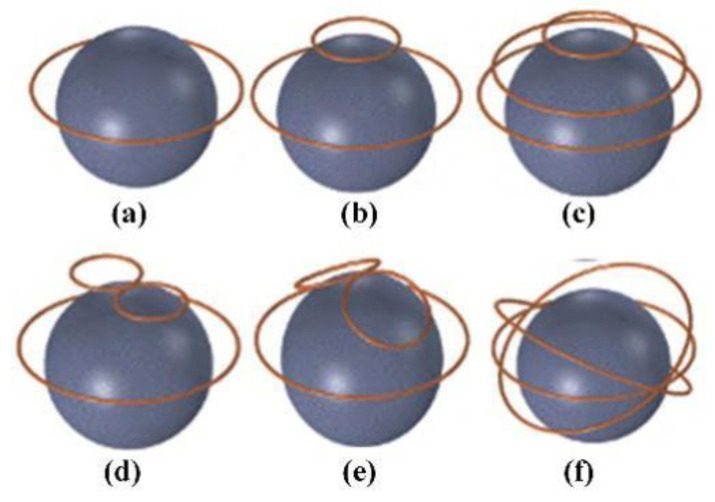
The structure of the Halo coil and related combinations. (**a**) Halo coil. (**b**) HCA coil. (**c**) HTC coil. (**d**) HFA coil. (**e**) HDA coil. (**f**) THC coil.

**Figure 10 biosensors-11-00225-f010:**
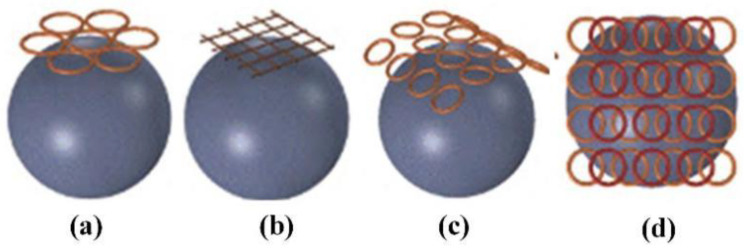
The structure of the TMS coil arrays. (**a**) Circular coil array. (**b**) Linear coil array. (**c**) Cap-formed coil array. (**d**) Multi-layer coil array.

**Figure 11 biosensors-11-00225-f011:**
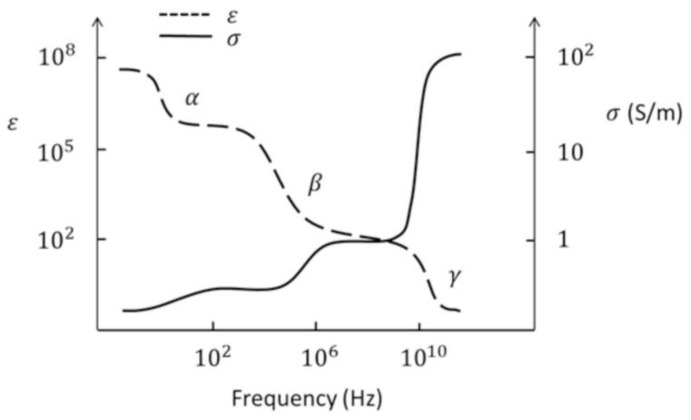
The dispersions of biological samples.

**Figure 12 biosensors-11-00225-f012:**
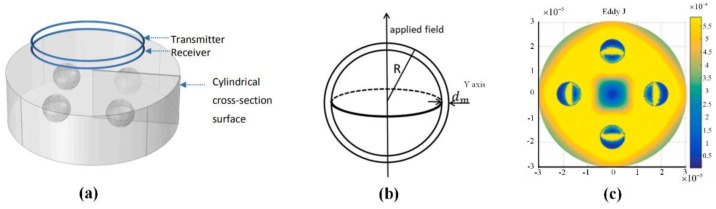
Three-dimension finite element model. (**a**) Measurement set-up. (**b**) Conductivity spectroscopy of fresh potato and defrosted potato. (**c**) Simulation model.

**Figure 13 biosensors-11-00225-f013:**
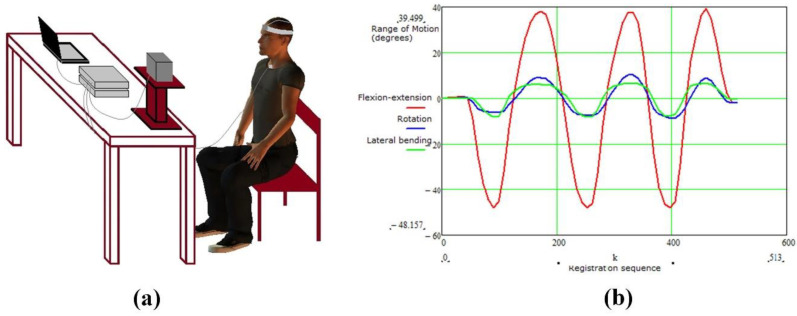
Three-dimensional electromagnetic tracking system. (**a**) Schematic diagram of detection device. (**b**) Movement measurement signal. Adapted with permission from ref. [107]. Copyright 2018 Elsevier.

**Figure 14 biosensors-11-00225-f014:**
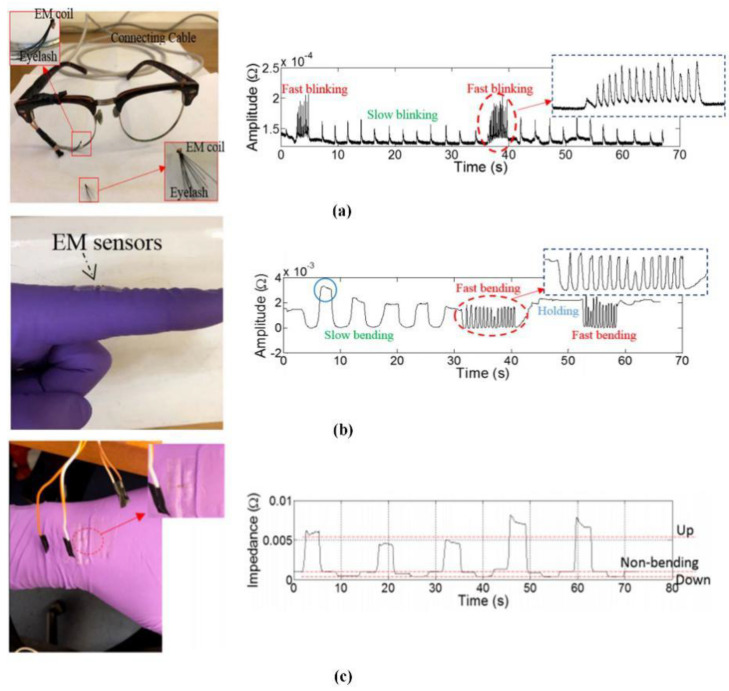
The biomechanical motion monitoring based on electromagnetic biosensors. (**a**) Wearable EM sensor. (**b**) Finger bending. (**c**) Wrist bending. Adapted with permission from ref. [108]. Copyright 2020 IEEE.

**Figure 15 biosensors-11-00225-f015:**
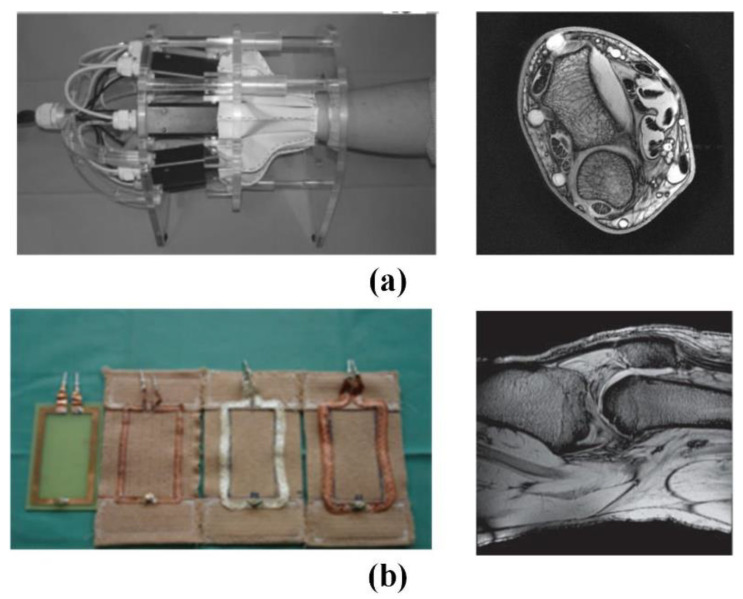
The adjustable mechanical structure for MRI. (**a**) Wrist coil and corresponding detection results. Adapted with permission from ref. [118]. Copyright 2009 Wiley-Liss. (**b**) Knee joint coil, consisting of the reference coil made from circuit and stretchable coils made from 5 mm, 7 mm, and 9 mm copper braids, and detection results. Adapted with permission from ref. [119]. Copyright 2012 Wiley-Liss.

**Figure 16 biosensors-11-00225-f016:**
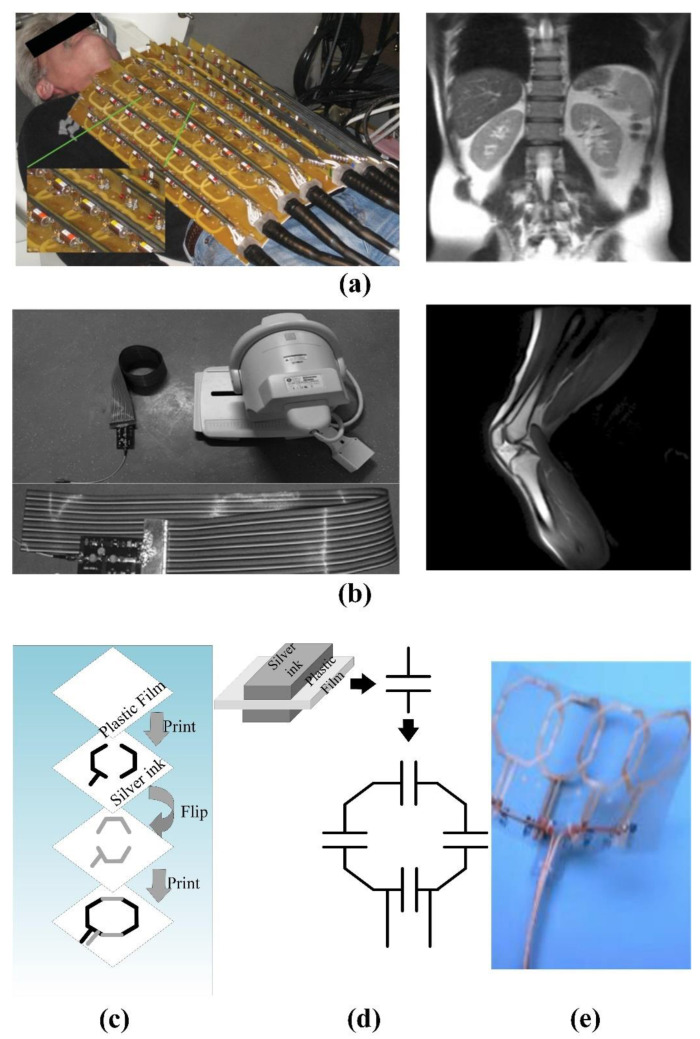
The wearable MRI sensor. (**a**) The 128-channel torso array coils and corresponding imaging results. Adapted with permission from ref. [117]. Copyright 2008 Wiley-Liss. (**b**) Lightness and softness coils and corresponding imaging results. Adapted with permission from ref. [120]. Copyright 2015 John Wiley & Sons. (**c**) Coil printing process flow. (**d**) Schematic of a printed coil. (**e**) Photograph of a printed coil.

**Figure 17 biosensors-11-00225-f017:**
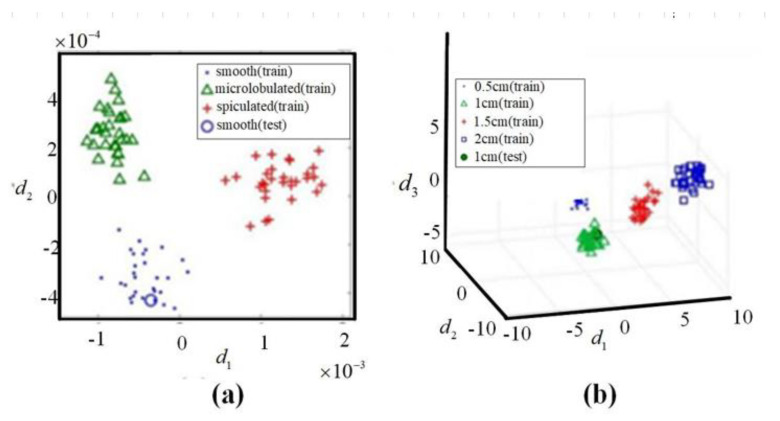
The classification results of the morphological characteristics of a breast tumor. (**a**) Shape. (**b**) Size.

**Figure 18 biosensors-11-00225-f018:**
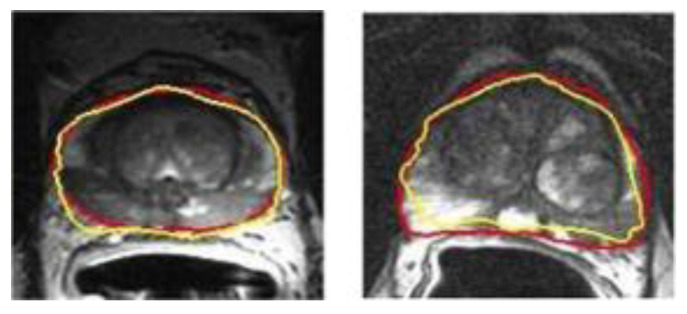
Segmentation results of glands and their boundaries. Red: ground truth; yellow: segmentation contour by the proposed algorithm. Adapted with permission from ref. [138]. Copyright 2018 Elsevier.

**Figure 19 biosensors-11-00225-f019:**
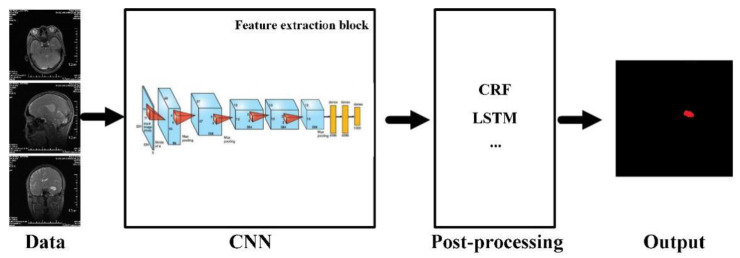
The basic method of MRI brain tumor segmentation based on CNN.

**Figure 20 biosensors-11-00225-f020:**
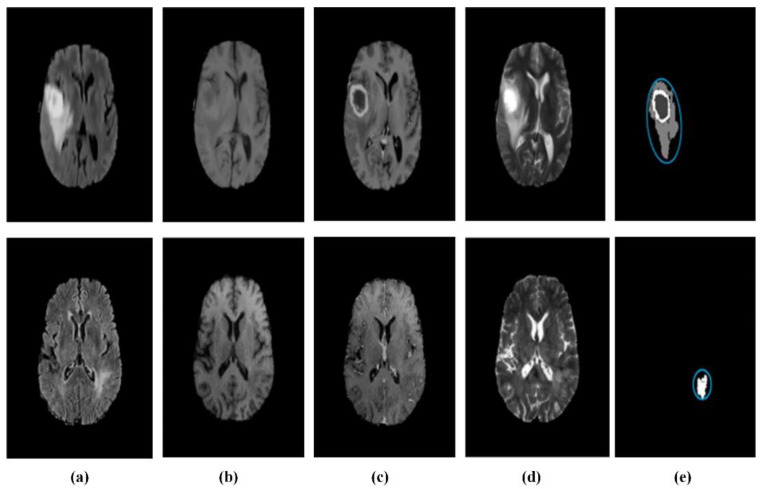
Brain tumor segmentations by multi-network. (**a**) Flair. (**b**) T1. (**c**) T1c. (**d**) T2. (**e**) Segmentation results. Adapted with permission from ref. [147]. Copyright 2018 Elsevier.

**Figure 21 biosensors-11-00225-f021:**
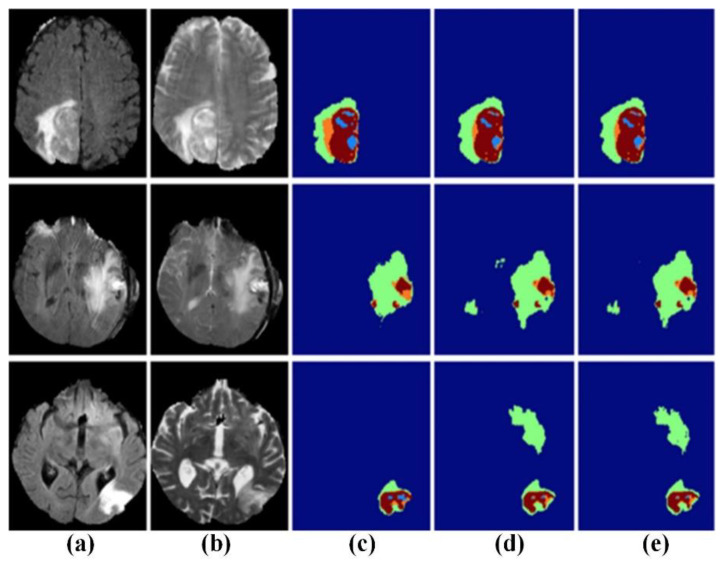
Brain tumor segmentations by 3D CNN. (**a**) Flair. (**b**) T2. (**c**) Manual. (**d**) Deep Medic. (**e**) Deep Medic + CRF.

**Figure 22 biosensors-11-00225-f022:**
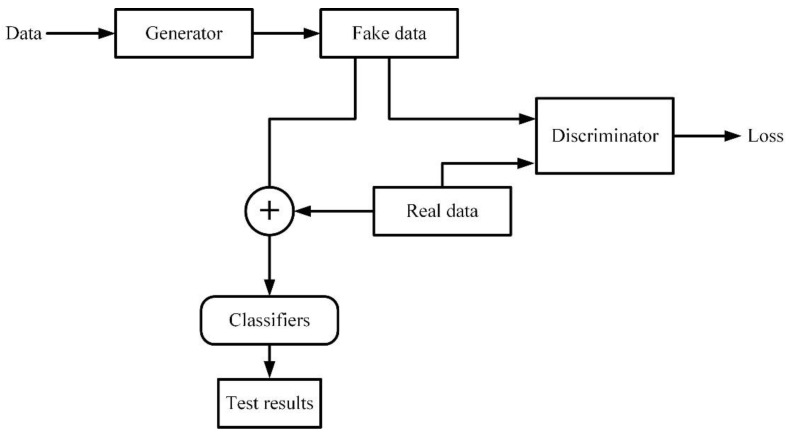
The MRI brain tumor segmentation based on GAN.

**Table 1 biosensors-11-00225-t001:** Electromagnetic biological effects and related applications.

Electromagnetic Biological Effect	Application	References
Thermal effects	Seed sterilization and inactivation	[15]
	Food storage	[16]
	High frequency electric knife	[17]
	Transcranial magnetic stimulation	[18]
	NMR	[19]
Non-thermal effects	Cardiac pacing	[20]
	Cardiac defibrillation	[21]
	Tumor treatment	[22]
Cumulative effects	Transcranial magnetic stimulation	[23]
	Fracture healing	[24]

**Table 2 biosensors-11-00225-t002:** Comparison of the thawing meat products under high-voltage electrostatic field.

Material	Sensor Structure Parameters	Thaw Effect	Reference
Beef	4 cm × 4 cm × 2 cm, Space of electrodes: 4 cm, 10 kv	The thawing time decreases as the number of electrodes increases	[39]
Tuna	2 cm × 4 cm × 4 cm, number of electrodes: 16, 5–14 kv	Thawing time is significantly reduced	[38]
Pork	5 cm × 5 cm × 1 cm, Space of electrodes: 5 cm, −10 kv	The pH and tenderness do not present obvious variation from normal air thawing	[40]

**Table 3 biosensors-11-00225-t003:** Characteristics of the Halo coil model.

Coil Model	Advantages	Disadvantages
Halo	Increase the electromagnetic penetration depth	The required the current is relatively large
HCA	Improve the penetration depth	Poor deep focality
HTC	Better deep focality	Compared with HCA, the strength of the induced electric field is reduced
HFA	The electric field strength and penetration depth increase on the side	The focality is poor and the attenuation rate is increased
HAD	Compared with HFA, the strength and penetration are larger	The focality at the gray and white areas is poor
THC	High flexibility	The superficial electric field strength is high

**Table 5 biosensors-11-00225-t005:** Evaluation results of MRI brain tumor segmentation based on the multi-CNN network.

Reference	Dataset	Evaluation Index DSC		
		Intact Tumor	Core Tumor	Enhance Tumor
[150]	BraTS 2015	0.92	0.84	0.77
[151]	BraTS 2015	0.90	0.76	0.73
[152]	BraTS 2017	0.72	0.83	0.81
[153]	BraTS 2013	0.80	0.67	0.85
[154]	BraTS 2013	0.88	0.79	0.83

## Data Availability

Not applicable.

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
