# Peer review of "Biomedical Applications of Electromagnetic Detection: A Brief Review"

_biosensors, 2021, doi:10.3390/bios11070225_

Round 1

Reviewer 1 Report

The work presented in the review paper is comprehensive, interesting, and well-suited for publication in the journal of biosensors. This is a review on the effects of electromagnetic biosensors, their mechanisms, and their applications at different frequencies, and the use of machine learning technology in electromagnetic medical images. This is a well-written review and I only have a few minor points for the authors.

  1. The axis labels in Figure 12c should be increased in font size. They are not very visible.
  2. In figure 16c, there are more sub-labels ‘a’, ‘b’, ‘c’, and ‘d’ which are not labelled. Please explain or label the smaller figures under 16c.
  3. In figure 15b, there are more sub-labels ‘a’, ‘b’, ‘c’, and ‘d’ which are not labelled. Please explain or label the smaller figures under 15b.
  4. In figure 20, there are 5 different sub-figures and one of them has red/green color pigments. Please explain what the different sub-figures represent exactly and explain the red/green stains on the image.
  5. A table should be included summarizing all the different electromagnetic biosensor effects, their applications, and some key references which illustrate these effects and applications. This would give readers a good overview and easy comprehension of a big review.

Author Response

Responses to Reviews

  1. The axis labels in Figure 12c should be increased in font size. They are not very visible.

Thank you for your suggestion. I have increased the font size of axis labels in Figure 12c.

  1. In figure 16c, there are more sub-labels ‘a’, ‘b’, ‘c’, and ‘d’ which are not labelled. Please explain or label the smaller figures under 16c.

Thank you for your suggestion. I have made corrections. They are label 16c, 16d and 16e respectively.

  1. In figure 15b, there are more sub-labels ‘a’, ‘b’, ‘c’, and ‘d’ which are not labelled. Please explain or label the smaller figures under 15b.

Thank you for your suggestion. They are knee joint coil consisted by the reference coil made from circuit and stretchable coils made from 5-mm, 7-mm, 9-mm copper braid, respectively. I have made caption to describe the content of figure 15b.

  1. In figure 20, there are 5 different sub-figures and one of them has red/green color pigments. Please explain what the different sub-figures represent exactly and explain the red/green stains on the image.

Thank you for your suggestion. The 5 different sub-figures are Flair, T1, T1c, T2 and segmentation results. The red and green represents ground truth and segmentation results.

  1. A table should be included summarizing all the different electromagnetic biosensor effects, their applications, and some key references which illustrate these effects and applications. This would give readers a good overview and easy comprehension of a big review.

Thank you for your suggestion. I have made a table to summarize the different electromagnetic biosensor effects and corresponding applications and references.

Reviewer 2 Report

Review of Electromagnetic Sensors for Biomedical Applications

While this review is interesting to read, it is difficult to see how it deals directly with electromagnetic sensors/biosensors. Furthermore, the focus of the review is unclear. The introduction section deals with biosensors and on line 37 the authors state ‘biosensors can be traced back to the 1960s when Professor Clark first proposed the concept of enzyme sensors, giving the impression that the remaining sections will focus on biosensors (a device that converts a biological response into a measurable signal). Instead, the review covers initially the biological effects of electromagnetic fields, and then covers several topics varying from food quality inspection, food sterilisation, plant cultivation, seed germination and growth, thawing technology, food preservation, NMR, breast cancer detection, brain stroke, artificial intelligence and machine learning. There is very little detail on the measured sensor signals.

This review tries to cover too many topics with a lack of focus, and as a result each section is described in very general terms with a lack of in-depth analysis. The 153 references are divided between several distinct topics, reflecting the lack of an in-depth discussion.

This review needs to be re-written, either change the title and introduction to match and reflect the content, e.g. Review on the applications of electromagnetic fields in ....Alternatively, focus on electromagnetic sensors, providing more details on the signals measured and how they can be used to quantify the parameter being measured.  Also, a more in-depth analysis is required.

Author Response

Thank you for your suggestion. I have modified our review. For one thing, I have change the title and introduction to match and reflect the content. The title of this paper has been revised the Biomedical Applications of Electromagnetic detection: A Brief Review. Moreover, the introduction is also revised and emphasizes the effect of electromagnetic fields on organism. For another, this review is added more details on the signals for electromagnetic sensors, which brings the intuitive presentation of the working effects of these sensors for readers.

Reviewer 3 Report

The article entitled “Review of Electromagnetic Sensors for Biomedical Applications” exploits application of electromagnetic biosensors using frequency as an evidence as well as electromagnetic biosensor cooperated with machine learning (ML) technology have been explained in clinical diagnosis because of its powerful feature extraction capabilities. The article is very well-written and reading is fluid and engaging. However, your title is rather general and doubts are raised if novelty exists. Please reinforce it. Compare for instance your work to the ones presented in articles with doi: 10.3390/s19071662, doi.org/10.3390/s21041109. Here are some of the comments and suggestions to the authors which have to be addressed before accepting for publication. 1. Most figures are taken from the other sources. Have the authors get the copy rights permission? If yes, then they should mention in the figure caption. 2. Figure 2, there should be clear scale bar of SEM images 3. Some figures should be changed with high resolution figures using TIFF file for easily and clearly readable for the readers 4. Authors should explain the future challenges or prospects with a news section in this review article. 5. MRI brain tumor segmentation based on CNN should be more explained 6. All figure captions need improvements. 7. The text of Figure 22 is un-readable. 8. I would like to encourage the authors to focus the last 5 years’ work. 9. When citing the authors work, it is appropriate to mention et al after the name of the author unless authored by a single person. For example, Li et al 10. Some citations in the reference section aren’t formatted properly. After corrections and additions, the manuscript can be published in biosensors Journal.

Author Response

1.Most figures are taken from the other sources. Have the authors get the copy rights permission? If yes, then they should mention in the figure caption.

Thank you for your suggestion. I have got the copy rights permission and mentioned in the figure caption.

2.Figure 2, there should be clear scale bar of SEM images.

I have modified the SEM images to make the scale bar clear.

3.Some figures should be changed with high resolution figures using TIFF file for easily and clearly readable for the readers.

Thank you for your suggestion. I have improved the clarity of the image to make it easier for readers to read.

4.Authors should explain the future challenges or prospects with a news section in this review article.

I have modified it according to your suggestion.

5.MRI brain tumor segmentation based on CNN should be more explained.

I have added explanations of MRI brain tumor segmentation based on CNN in the manuscript.

6.All figure captions need improvements.

I have modified it according to your suggestion.

7.The text of Figure 22 is un-readable.

Thank you for your suggestion. I have modified it according to your suggestion.

8.I would like to encourage the authors to focus the last 5 years’ work.

Thank you for your suggestion. I have added some works on electromagnetic biosensors in the last five years.

9.When citing the authors work, it is appropriate to mention et al after the name of the author unless authored by a single person. For example, Li et al

I have made corrections.

10.Some citations in the reference section aren’t formatted properly.

Thank you for your suggestion. I have added some works on electromagnetic biosensors in the last five years.

Round 2

Reviewer 2 Report

The authors have modified the title and introduction section to reflect more accurately of what is reviewed in the paper. The topic is interesting and more details are now provided in each section. Therefore I can now recommend that this manuscript be accepted for publication.